# Controllable Carbonization of Plastic Waste into Three-Dimensional Porous Carbon Nanosheets by Combined Catalyst for High Performance Capacitor

**DOI:** 10.3390/nano10061097

**Published:** 2020-06-02

**Authors:** Xueying Mu, Yunhui Li, Xiaoguang Liu, Changde Ma, Hanqing Jiang, Jiayi Zhu, Xuecheng Chen, Tao Tang, Ewa Mijowska

**Affiliations:** 1School of Chemistry and Environmental Engineering, Changchun University of Science and Technology, Changchun 130022, China; xymu@ciac.ac.cn; 2State Key Laboratory of Polymer Physics and Chemistry, Changchun Institute of Applied Chemistry, Chinese Academy of Sciences, Changchun 130022, China; cdma@ciac.ac.cn (C.M.); hqjiang@ciac.ac.cn (H.J.); 3Faculty of Chemical Technology and Engineering, West Pomeranian University of Technology, Piastow Ave. 42, 71-065 Szczecin, Poland; Xiaoguang.Liu@zut.edu.pl (X.L.); emijowska@zut.edu.pl (E.M.); 4State Key Laboratory of Environment-friendly Energy Materials, School of Science, Southwest University of Science and Technology, Mianyang 621010, China; jyzhu@swust.edu.cn

**Keywords:** PET, waste plastic, carbonization, energy storage

## Abstract

Polyethylene terephthalate (PET) plastic has been extensively used in our social life, but its poor biodegradability has led to serious environmental pollution and aroused worldwide concern. Up to now, various strategies have been proposed to address the issue, yet such strategies remain seriously impeded by many obstacles. Herein, waste PET plastic was selectively carbonized into three-dimensional (3D) porous carbon nanosheets (PCS) with high yield of 36.4 wt%, to be further hybridized with MnO_2_ nanoflakes to form PCS-MnO_2_ composites. Due to the introduction of an appropriate amount of MnO_2_ nanoflakes, the resulting PCS-MnO_2_ composite exhibited a specific capacitance of 210.5 F g^−1^ as well as a high areal capacitance of 0.33 F m^−2^. Furthermore, the PCS-MnO_2_ composite also showed excellent cycle stability (90.1% capacitance retention over 5000 cycles under a current density of 10 A g^−1^). The present study paved an avenue for the highly efficient recycling of PET waste into high value-added products (PCSs) for electrochemical energy storage.

## 1. Introduction

Due to its light weight and outstanding mechanical properties, plastic has been widely used in many fields. However, with the ever-increasing consumption of plastic products, it has flooded almost every corner of peoples’ lives, and its poor biodegradability has resulted in serious environment pollution [1,2]. Currently, landfill and incineration are the most widely adopted strategies for the treatment of plastics wastes [3,4,5]. Nevertheless, in the long term, the decomposition of plastic waste buries underground will lead to soil and underground water contamination and harmful gases will be released into the air. However, the inappropriate treatment of plastic waste will not only cause resource waste, but also induce serious environmental pollution [6]. Although many strategies have been proposed to treat and reuse the plastic waste, it is still a great challenge to recycle the plastic waste at large scale [7]. Among various commonly used plastics, polyethylene terephthalate (PET) is widely used in our social life. For example, in China, all plastic bottles are made from PET and up to 3 million tons of PET plastic bottles are consumed every year. Therefore, developing a method for efficient recycling of PET waste plastics has a significant impact on resources and the environment. Considering the high carbon content of PET (62.5 wt%), the transformation of PET waste into carbon materials is regarded as one of the most potentially beneficial methods due to the wide applications of carbon materials in various research fields [8]. However, the native low carbonization yield of PET seriously prohibits the implementation of this strategy as only 20 wt% of carbon material can be recycled. This is much lower than expected. Therefore, it is still a challenge to efficiently carbonize PET with high yield [9]. More importantly, it is hard to achieve controllable carbonization of PET into pre-designed nanostructures with determined physical and chemical properties [10,11].

Up to now, many research works have been published on recycling of PET waste [12,13,14,15]. For example, Elessawy et al. mixed PET waste with urea to react in autoclave reactor to prepare 3D sponge nitrogen-doped graphene [16]. The prepared N-doped graphene shows excellent specific capacitance and energy density. Kamali et al. mixed PET with NaCl with subsequent heat treatment, in which molten salt acted as a graphitization medium to improve the quality of PET plastic-derived carbon [17]. As an important energy storage device, the supercapacitor has drawn huge attention in the scientific community due to its high power density, long cycle life, and wide operating temperature range [18,19,20]. Although many achievements have been made, the carbonization yield is still much lower than expected. As the most essential component of the supercapacitor, the electrode material mainly determines the electrochemical performance, the exploitation of new advanced electrode materials has become the research hotspot in recent years. In general, electrode materials can be divided into three categories: carbon materials, metal oxides, and polymers [21]. Because of the excellent thermal/chemical stability, excellent conductivity, and tunable surface area, carbon materials are the most extensively studied electrode materials so far [22,23,24,25]. According to the molecular structure, the carbon content of waste PET plastic is as high as 62.5 wt%. Therefore, it is highly desired to transform PET waste into a carbon material for electrodes, which will not only reduce the environmental burden, but also provide advanced electrode materials for energy storage. It is of considerable significance to develop an efficient, time-saving, and cost-effective strategy to recycle PET waste, and more importantly, achieving controllable and high yield carbonization of PET waste simultaneously.

According to previous research, it is feasible that MgO can be used as a hard template to prepare two-dimensional carbon nanosheets, but stacking is likely to occur between carbon flakes due to π-π interaction. When cobalt acetylacetonate (III) (Co(acac)_3_) is fully mixed with MgO flakes to form a hybrid catalyst, the resulting three-dimensional structure will not only stop the re-stacking of carbon nanosheets, but also facilitate the ion transportation within the carbon framework and further improve capacitor performance. In this contribution, waste PET bottles were used as carbon source, the hybrid MgO/Co(acac)_3_ was used as a combined catalyst, and the porous carbon nanosheets (PCS) were selectively prepared by catalytic carbonization of PET waste. The yield of PCS was up to 36.4 wt%, which is much higher than that of neat PET carbonization product (CP) yield of 22 wt%, demonstrating the high catalytic activity of the prepared combined catalyst. MnO_2_ nanoflakes were further loaded on the resulting PCSs by redox reaction to obtain hybrid PCS-MnO_2_ composite, which introduced the pseudo-capacitance and further improved the capacitive performance of the hybrid material. More importantly, the PCS-MnO_2_ showed an ultrahigh areal capacitance of 0.33 F m^−2^, as well as long cycle stability [26].

## 2. Materials and Methods

### 2.1. Materials

MgO was purchased from China Guangdong Xilong Science Co., Ltd., and Co(acac)_3_ (98%) was obtained from Aladdin Biochemical Technology Co., Ltd. (Shanghai, China). KMnO_4_ was purchased from Beijing Chemical Plant (Beijing, China). The waste PET plastic used was a recycled waste plastic bottle. Anhydrous ethanol and HCl acid were purchased from Beijing Chemical Plant (Beijing, China), without any further purification.

### 2.2. Preparation of Porous Carbon Nanosheet (PCS)

Neat PET carbonization product (CP) was obtained by direct carbonization of waste PET in a reaction kettle at 700 °C. For the preparation of PCS, in the typical synthesis, MgO and Co(acac)_3_ were put into a ball mill with a mass ratio of 2:1 to get a combined catalyst (MgO/Co(acac)_3_). The waste PET was firstly cut into small fragments and mixed with the combined catalyst at a mass ratio of 1:2, followed by introducing into an autoclave to react at 700 °C for 1 h. Subsequently, the carbon product was taken out and refluxed in 1M HCl acid solution at 100 °C for 12 h to remove MgO and cobalt oxide from the carbon sample. In the next step, the mixture was filtered to get the PCSs sample. Finally, the PCSs were dried in vacuum at 80 °C prior to the electrochemical test.

### 2.3. Preparation of Porous Carbon Nanosheet (PCS-MnO_2_)

To load MnO_2_ nanoflakes on PCSs, in a typical synthesis, 25*x* mg (*x* = 1, 2, 4) of KMnO_4_ was dissolved in 500 mL deionized water, and the solution was heated at 80 °C in an oil bath. Then 100 mg of the as-prepared carbon material (PCS) was added and stirred gently for 1 h. After cooling to room temperature, the PCS-MnO_2_ composite was separated from the mixture and dried in vacuum for further use. The final composite product was denoted as PCS-MnO_2_-*x* (*x* = 1, 2, 4).

### 2.4. Characterization

The morphology of the carbon nanomaterial was measured by a field emission scanning electron microscope (SEM, XL30ESEM-FEG) of Japan JEOL Company at an acceleration voltage of 20 kV, and the microscopic structure of the carbon nanomaterial was tested by a transmission electron microscope (TEM, JEM-1011) of Japan JEOL Company at an acceleration voltage of 100 kV. The phase composition of the carbon nanomaterial was determined by X-ray diffraction (XRD, D8 Advance X-ray diffractometer) of German Bruker Company, and the purity and thermal stability of carbon material was analyzed by a thermogravimetric analyzer (TA Instruments SDT Q600) of American TA company. The elemental composition and contents of the materials were determined by X-ray photoelectron spectroscopy (VG ESCALAB MK II) of American Thermo Scientific company at 10.0 kV and 10 mA Al Kα radiation. The N_2_ adsorption/desorption isotherms were obtained at liquid nitrogen temperature (77 K) on the Micromeritics ASAP 2010 M of American Micromeritics company. The density functional theory (DFT) and Brunauer–Emmett–Teller (BET) methods were used to calculate the specific surface area and pore size distribution.

### 2.5. Electrochemical Test

To prepare the working electrode, in a typical synthesis, the active material, conductive carbon black, and 5 wt% of polytetrafluoroethylene (PTFE) solution were mixed together to form a paste with a mass ratio of 8:1:1, then the paste was uniformly coated on nickel foam (area: 1 cm^2^), the loading amount of the active material of the obtained electrode was approximately 4 mg cm^−2^. And then placed in a vacuum drying oven at 80 °C for 12 h. Then, a roller press was applied to press the electrode sheet to a thickness of 0.5 mm. A platinum plate was used as a counter electrode for a three-electrode test system, and 6 mol L^−1^ of KOH was used as the electrolyte. After the electrode sheet was soaked in the electrolyte for 12 h, the electrochemical tests were carried out on the electrochemical workstation (CHI 660E). Cyclic voltammetry (CV) was tested at a voltage window of −1~0 V with a scan rate of 1~200 mV s^−1^. The galvanostatic charge-discharge (GCD) test was carried out under the test voltage window of −1~0 V and with a current density of 0.5~20 A g^−1^. The electrochemical impedance test is in range of 0.01 Hz~100 kHz. In the three-electrode system, the gravimetric specific capacitance and the areal specific capacitance is calculated from the following equations: (1)$$C_{wt(CV)}=\frac{1}{2ms\left(V_{b}-V_{a}\right)}\int_{v_{a}}^{v_{b}}IdV$$
(2)$$C_{wt(GCD)}=\frac{I*\Delta t}{m\left(V_{b}-V_{a}\right)}$$
(3)$$C_{areal}=\frac{C_{wt\left(CV\right)}}{S_{BET}}$$
where *C_wt(GCD)_* (F g^−1^) and *C_wt(CV)_* (F g^−1^) are the gravimetric specific capacitance (*C*) calculated from GCD and CV results, *I* (A) is the current response, *V_b_* and *V_a_* are the high and low potential limits, *m* represents the carbon material mass, *s* (mV s^−1^) is the scanning rate, and *C_areal_* (F m^−2^) is the areal specific capacitance, which is obtained by dividing the gravimetric specific capacitance by the specific surface area of carbon. *S_BET_* (m^2^ g^−1^) is the specific surface area calculated from the N_2_ adsorption and desorption (77 K) isotherm.

## 3. Results

The synthetic route for the preparation of PCS-MnO_2_ composites is presented in Figure 1a. At the elevated temperature to 700 °C in the reactor, PET firstly decomposed into small organic molecules such as methane, ethylene, and other aromatic hydrocarbons [27]. At the same time, CoO nanoparticles were formed by the decomposition of Co(acac)_3_ and distributed on MgO flakes evenly. Based on the decomposition–diffusion–precipitation mechanism [28], the produced small organic molecules began to deposit on the surface of CoO/MgO undergoing a dehydrogenation and aromatization reaction to form carbon layer coated on CoO/MgO [29]. After the removal of MgO and cobalt oxide species, the PCSs were finally obtained. Notably, these curved carbon nanosheets can effectively prevent carbon sheets from re-stacking process. MnO_2_ nanoflakes were subsequently deposited on PCSs through the redox reaction to form PCS-MnO_2_ composite [30]. The corresponding chemical reaction is shown below [31]: 4MnO_4_^−^ + 3C + 3H_2_O → 4MnO_2_ + CO_3_^2−^ + 2HCO_3_^−^

The SEM and TEM were measured to investigate the microstructure of the obtained samples at different stages. Figure 1b, f presents the TEM and SEM images of the prepared PCS. They reveal a 3D structured porous carbon sheet with a thickness of ca. 20 nm. Figure 1c–e exhibit TEM images of PCS-MnO_2_ hybrid composite. It is easy to observe that the morphology of the PCS-MnO_2_ composite are the flake MnO_2_ with a diameter of ca. 15 nm attached to the surface of the PCS. With the increase of the amount of KMnO_4_ added, the amount of flake MnO_2_ loaded on the PCS surface gradually enhanced from PCS-MnO_2_-1 to PCS-MnO_2_-4, and the thickness of MnO_2_ also increased slightly. Figure 1g shows the SEM image of PCS-MnO_2_-2 with the inset revealing MnO_2_ nanoflakes as indicated by red circle. Appendix A presents EDX mapping of PCS-MnO_2_-2. It is found that C, O, and Mn elements are evenly distributed in the measured sample.

Catalytic efficiency is one of the most important parameters to influence the applicability of the tested catalyst for carbonization of plastic waste. The obtained carbon product fabricated without adding any catalyst is in the form of carbon block (Figure 2a). The carbonization yield of this sample is only 22 wt% (see Figure 2d). When PET waste was mixed with MgO flakes, the carbonization yield increased to 25.1 wt%, slightly higher than PET itself. The morphology of this sample was in the form of flake structured carbon derivated from MgO template (Figure 2b). When only Co(acac)_3_ was introduced to PET waste, the carbonization yield increased to 26 wt%, demonstrating slightly higher catalytic activity of Co(acac)_3_ than MgO and PET itself. The morphology of this sample is depicted in Figure 2c. It shows core/shell structured carbon spheres. The size of carbon spheres is in the range of 20–50 nm. However, when Co(acac)_3_ and MgO were introduced together to form a combined catalyst, the carbonization efficiency was dramatically enhanced to 36.4 wt%. This is much higher than MgO and Co(acac)_3_ itself, further demonstrating the positive synergistic effect between Co(acac)_3_ and MgO. The carbon product is in the form of folded flakes with rich mesopores, as shown in Figure 1b.

The phase composition and crystallinity of carbon samples at different stages were determined by X-ray diffraction (XRD). As observed in Figure 3a, the typical (200) peak at 43° and (220) peak at 62° are assigned to MgO (JCPDS 77-2179), which demonstrate the existence of MgO template. Besides, there are typical peaks corresponding to CoO from decomposition of Co(acac)_3_ (JCPDS 75-0418) in the pattern. It is believed that the formed CoO with MgO synergistically promote the dehydrogenation and carbonization of the polymer when considering the carbonization yield [29,32]. In the XRD spectrum of PCS there are only two strong peaks at 26° and 43°corresponded to (002) and (101) crystal planes of hexagonal graphite [33]. In PCS-MnO_2_-2, except for the peak at 26°, there are typical δ-MnO_2_ (JCPDS 42-1317) peaks, which have been marked in the curves, indicating the existence of MnO_2_ in the composite [34].

TGA was carried out to determine the carbon content of the tested samples. All of the materials were tested under air atmosphere. As shown in Appendix A, the amount of residual carbonized initial product is 74.4%, which indicates the amount of catalyst in the carbonized product. At 800 °C, the residue of PCS sample is 0%, implying that MgO/CoO has been completely removed from PCS. Three PCS-MnO_2_ samples with different amounts of MnO_2_ are clearly shown in Appendix A. It is worth noting that the sample of PCS-MnO_2_ begins to lose weight before 300 °C, which is due to the release of water trapped in the sample [35]. The weight loss in the range of 300~450 °C is due to the oxidation of PCS. In addition, due to chemical reaction of MnO_2_ to Mn_2_O_3_, a weak weight loss is also found after 450 °C [36]. Figure 3b presents the Raman spectra of CP, PCS and PCS-MnO_2_-2, respectively. All of them show two typical peaks at 1340 cm^−1^ and 1590 cm^−1^. The D band at 1340 cm^−1^ is related to the disordered vibration peak of amorphous carbon, and the G band at 1590 cm^−1^ corresponds to the ordered carbon [37]. The ratio of I_D_/I_G_ is used to estimate the graphitization degree of the carbon material. The higher I_D_/I_G_ ratio implies the lower graphitization degree. According to calculation, the I_D_/I_G_ value of CP and PCS is 1.12 and 0.99, respectively. The lower I_D_/I_G_ value indicates that the porous structure of PCS has a higher graphitization degree with respect to CP due to the existence of CoO inducing a higher graphitization degree. After loading of MnO_2_ on PCS, the I_D_/I_G_ value of PCS-MnO_2_-2 slightly increases to 1.06, implying that the graphitization degree decreased. At the same time, two new weak peaks at 569 cm^−1^ and 641 cm^−1^ appear which correspond to δ-MnO_2_ in PCS-MnO_2_-2. This is consistent with the XRD result [38].

XPS analysis was used to investigate the surface chemical state of each element in the carbon samples. Appendix A shows XPS spectra of PCS-before HCl treatment, PCS, PCS-MnO_2_-2 (a), and the narrow spectra of Mn_2p_ peaks of the PCS-MnO_2_-2 (b). Appendix A shows high-resolution spectrum of Mn2p with the peaks at 653.8 eV and 642.2 eV corresponding to the Mn2p_1/2_ and Mn2p_3/2_, respectively. Moreover, the energy band of 11.6 eV between Mn2p peaks indicates the presence of δ-MnO_2_, which is also in agreement with the XRD and EDX mapping results [39]. Figure 4a displays the N_2_ adsorption/desorption isotherms of PCS and PCS-MnO_2_ composites, respectively. Based on IUPAC specifications, lower N_2_ adsorption at low relative pressure (P/P_0_ < 0.05) and adsorption at high relative pressure (P/P_0_ > 0.9) meet IV adsorption isotherm for two curves, indicating the abundant micropores and mesopores existed [40]. Meanwhile, the hysteresis loop in the curves also indicates the existence of mesopores. The above results are consistent with the pore size distribution analysis (Figure 4b). Figure 4b shows the numerous mesopores that exist in the PCS, which is beneficial for the ions and electrolytes free transportation within the pores. The specific surface area of PCS and PCS-MnO_2_-*x* (*x* = 1, 2, 4) is reduced from 561 m^2^ g^−1^ to 453 m^2^ g^−1^, and the total pore volumes are 2.4 cm^3^ g^−1^, 2.32 cm^3^ g^−1^, 2.16 cm^3^ g^−1^, and 1.86 cm^3^ g^−1^, respectively. This means that the specific surface area and pore volume of PCS-MnO_2_ composites decrease along with the increase in the amount of MnO_2_ loading. An appropriate amount of MnO_2_ loaded on PCS can not only introduce higher pseudo-capacitance to the material but also maintain a larger specific surface area and suitable pore structure, which endows the PCS-MnO_2_ composite based electrode with better electrochemical performance. 

A three-electrode system was first applied to investigate the electrochemical properties of the prepared carbon samples. In Figure 5a, the cyclic voltammetry (CV) curves of PCS and PCS-MnO_2_ composites at 1 mV s^−1^ are shown. The rectangular-like CV curve of PCS proves its typical electric-double-layer behavior. For PCS-MnO_2_ composites, there are two obvious reduction peaks from −0.4 V to −0.1 V, and another two obvious oxidation peaks in the range of −0.7 V and −0.4 V. According to previous reports, the reduction peak around −0.4 V and −0.1 V is due to the reversible redox reaction between Mn (IV) and Mn (III). While another two oxidation peaks are attributed to the oxidation reaction between Mn (III) and Mn (II) [41]. In addition, the CV integral area of PCS-MnO_2_-2 is much larger than that of PCS and other PCS-MnO_2_ composites. The gravimetric capacitance reaches up to 193.8 F g^−1^ at 1 mV s^−1^, which is demonstrating that appropriate MnO_2_ loading can introduce pseudo-capacitance while maintaining a large specific surface area and suitable pore structure to improve the electrochemical performance of electrode materials. The corresponding GCD test at a current density of 1 A g^−1^ is presented in Figure 5b. Similarly, nearly triangle shape of PCS illustrates the electrochemical behavior of the electric double layer. For PCS-MnO_2_ composites, there are two potential plateaus at −0.4~−0.1 V and −0.7~−0.4 V, which are consistent with the CV curves and correspond to reversible redox reaction of the transition of Mn (IV) and Mn (III), and oxidation reactions of the transition of Mn (III) and Mn (II). Among these materials, PCS-MnO_2_-2 has the longest discharge time. As the loading of MnO_2_ increases, the discharge time of the PCS-MnO_2_ composites firstly increases and then decreases. This is due to the accumulation of MnO_2_ in PCS-MnO_2_-2, which reduces the utilization of MnO_2_ and the ion transportation rate. The CV curves of PCS-MnO_2_-2 with the scan rate in the range of 1~200 mV s^−1^ are presented in Figure 5c. The shape of the test curve remains the rectangular shape as the scan rate increases. Because of a larger specific area and suitable pore structure of PCS-MnO_2_-2 composite, it guarantees the channel opened and facilitates fast ions diffusion [42]. Figure 5d shows the GCD results of PCS-MnO_2_-2 at a current density of 1~20 A g^−1^. The gravimetric capacitance is as high as 210.5 F g^−1^ at 0.5 A g^−1^. At high current density, PCS-MnO_2_-2 still shows quasi-triangular shape, which illustrates that PCS-MnO_2_-2 has a good rate performance. Figure 5e shows the specific capacitance of PCS and PCS-MnO_2_ composites at 1~20 A g^−1^. Under different current densities, the specific capacitance of PCS-MnO_2_-2 always remains the highest in comparison to other electrode materials. It means that PCS-MnO_2_-2 contains optimal MnO_2_ loading which maximizes the utilization of MnO_2_ while maintaining an appropriate pore structure. These features allowed to provide an excellent electrochemical performance of this sample. Figure 5f shows the areal capacitance of graphene, PCS, and PCS-MnO_2_-2 at 1 mV s^−1^. It must be noted that the areal capacitance of PCS-MnO_2_-2 is as high as 0.33 F m^−2^, which is much higher than that of graphene (0.21 F m^−2^) due to the introduction of pseudocapacitive reactions.

The Nyquist plots were used to investigate the kinetic characteristics of ion diffusion in the tested electrode materials. As shown in Figure 6a, there are three parts in the above Nyquist plots, the intersection with the real axis in the high-frequency region corresponding to combined series resistance (*R*s) of the materials. For PCS and three PCS-MnO_2_ composites, the Rs values are 2.08 ohm, 2.23 ohm, 2.49 ohm and 2.76 ohm, indicating that when more MnO_2_ is loaded on PCS, the intrinsic resistance correspondingly increases. In the mid-frequency region, the diameter of semicircle indicates the charge-transfer resistance (*R*ct) of the electrode materials [43]. The *R*ct of PCS and PCS-MnO_2_ composites are 1.07 ohm, 1.38 ohm, 1.6 ohm, and 1.73 ohm, respectively. The increased amount of MnO_2_ on PCS reduces the pore width of the composite materials, and enlarges the distance of ion transportation, resulting in higher charge transportation resistance. In the low-frequency region, the curves become an inclined line and nearly vertical line indicates the good ion diffusion ability of electrode materials. To measure the ion diffusion resistance, the Warburg coefficient s (ohm s^−1/2^) can be extracted by fitting the real part of impedance (Z′) versus the −1/2 power of the angular frequency (w^−1/2^) in the low frequency range (Figure 6b) [44]. The slope of the fitted line equals the Warburg coefficient (s). The Warburg coefficient values of PCS and PCS-MnO_2_ composites are 0.33 ohm s^−1/2^, 0.69 ohm s^−1/2^, 1.35 ohm s^−1/2^, and 1.97 ohm s^−1/2^, respectively. It is known that the Warburg coefficient is inversely proportional to the ion diffusion coefficient [45]. Therefore, it can be inferred from the Warburg coefficient and the change trend that abundant mesopores are beneficial for ion diffusion, and as the amount of loaded MnO_2_ increases, the difficulty of ion diffusion increases. Therefore, PCS-MnO_2_-2 loaded with a suitable amount of MnO_2_, can maintain a suitable porous characteristic while introducing pseudo-capacitance and displays high capacitances and excellent rate capabilities. Figure 6c shows that the specific capacitance still has a retention rate of 90.1% over 5000 times GCD cycles at a current density of 10 A g^−1^, the value is higher than those of PET-derived carbon materials previously reported (Appendix A) [46,47], which demonstrates the excellent cyclic stability of PCS-MnO_2_-2 composite.

## 4. Conclusions

In summary, with the use of MgO/Co(acac)_3_ as a template and catalyst, waste PET plastic was selectively carbonized into 3D porous carbon nanosheets with a high yield (36.4 wt%). After loading uniform MnO_2_ nanoflakes on carbon nanosheets through a redox reaction, the as-prepared PCS-MnO_2_-2 composite exhibited excellent capacitive performance in a supercapacitor. Due to the high specific surface area, appropriate pore size distribution, and uniformly distributed MnO_2_ active sites, the PCS-MnO_2_-2 composite delivered a gravimetric capacitance of 210.5 F g^−1^ and areal capacitance of 0.33 F m^−2^, as well as excellent cycle stability. The present work demonstrates a strategy for the one-step carbonization of PET waste plastics into PCSs for energy storage, achieving the goal of recycling plastic waste into a high value-added product.

## Figures and Tables

**Figure 1 nanomaterials-10-01097-f001:**
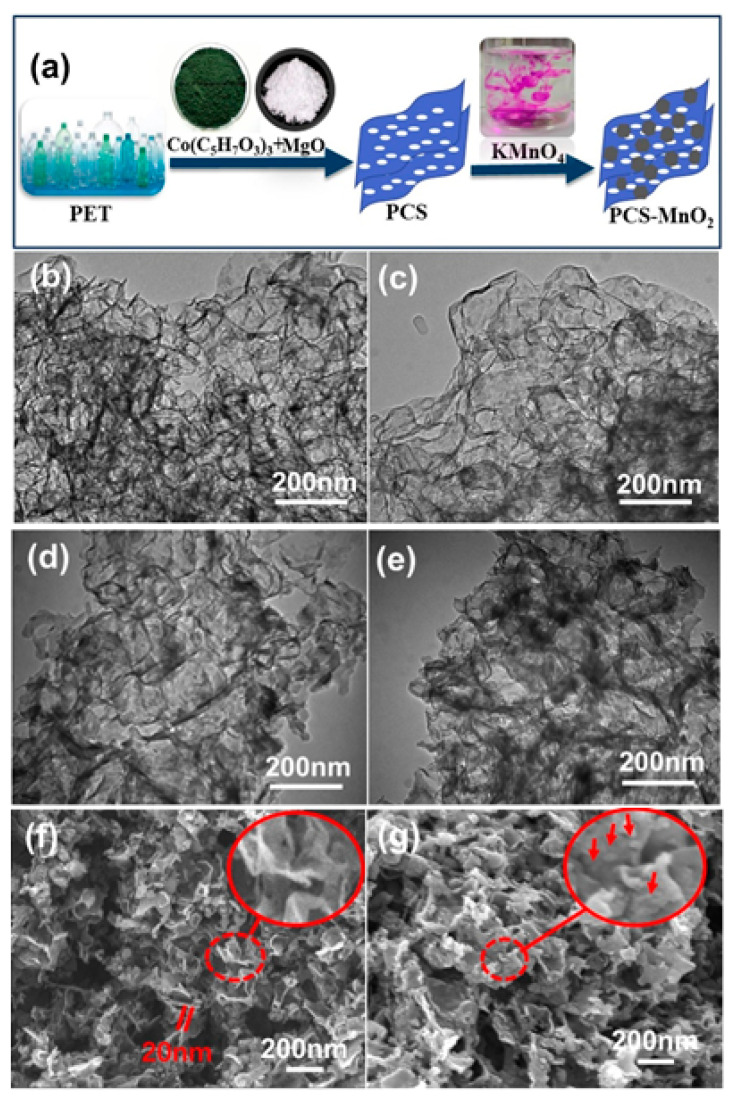
(**a**) Schematic illustration for the synthesis of PET waste derived PCS-MnO_2_. TEM images of (**b**) PCS (**c**) PCS-MnO_2_-1 (**d**) PCS-MnO_2_-2 (**e**) PCS-MnO_2_-4; SEM image of (**f**) PCS (**g**) PCS-MnO_2_-2.

**Figure 2 nanomaterials-10-01097-f002:**
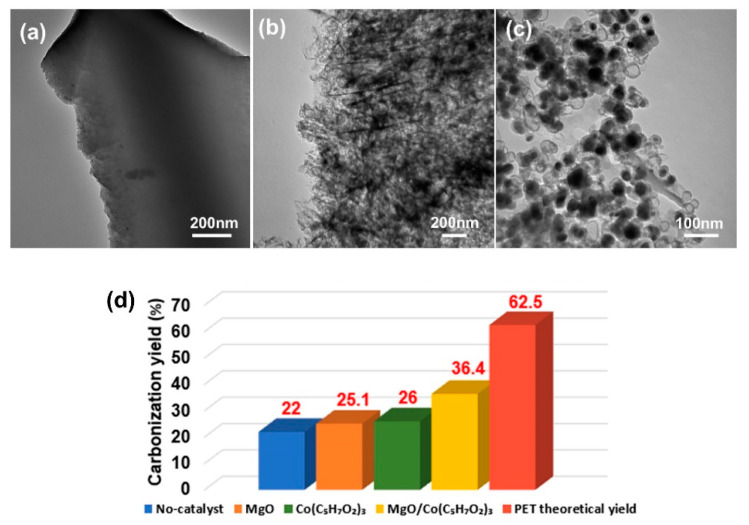
TEM images of (**a**) Carbon prodcut from neat PET, (**b**) Carbon sample prepared from PET/MgO, (**c**) Carbon sample from PET/Co(acac)_3_ and (**d**) the corresponding carbon yields from different catalyst.

**Figure 3 nanomaterials-10-01097-f003:**
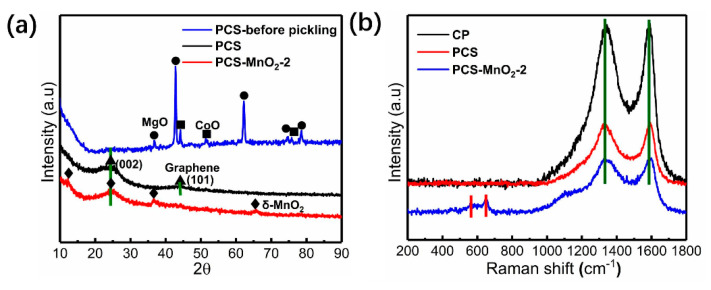
(**a**) XRD patterns of carbon product prepared from directly carbonized from Co(acac)_3_/MgO/PET composite, PCS and PCS-MnO_2_-2. (**b**) Raman spectra of CP, PCS and PCS-MnO_2_-2.

**Figure 4 nanomaterials-10-01097-f004:**
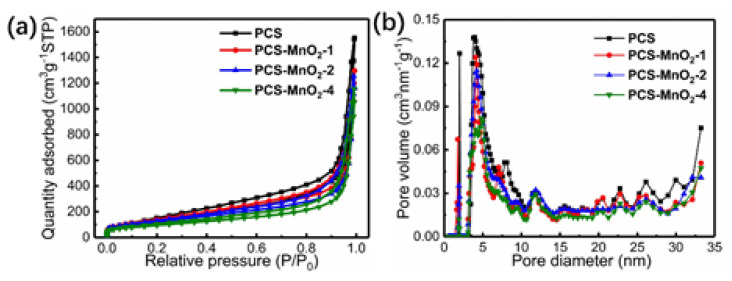
(**a**) N_2_ adsorption/desorption isotherms and (**b**) Pore size distributions of PCS and PCS-MnO_2_ composites, respectively.

**Figure 5 nanomaterials-10-01097-f005:**
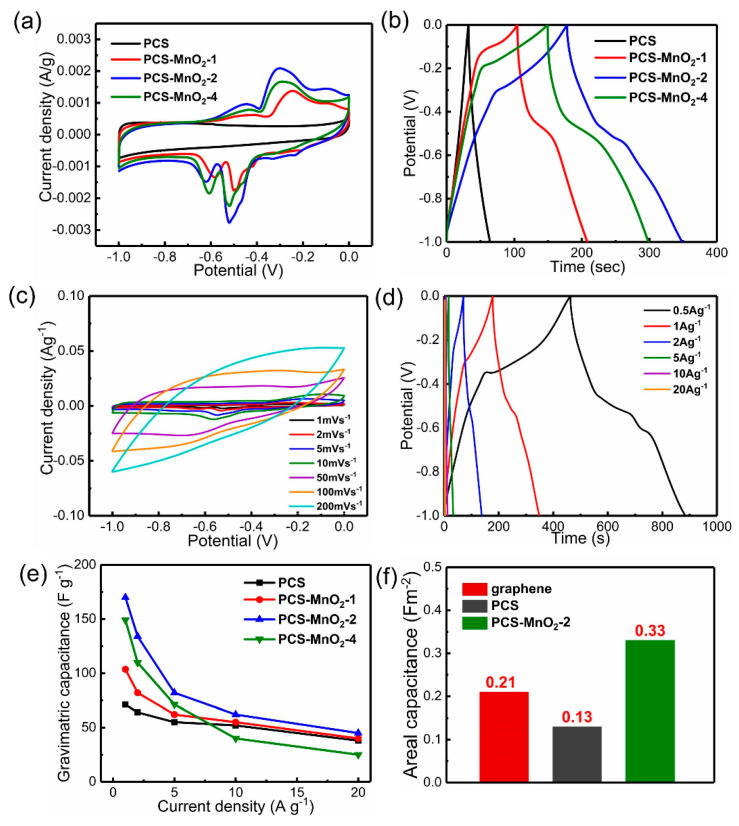
Electrochemical performance of PCS, PCS-MnO_2_ composites in 6 M KOH electrolyte in the three-electrode test system: (**a**) CV curves tested at a scan rate of 1 mV s^−1^, (**b**) GCD curves tested at the current density of 1 A g^−1^. (**c**) CV curves of PCS-MnO_2_-2 at the scan rate in the range of 1~200 mV s^−1^, (**d**) GCD curves of PCS-MnO_2_ at the current density in the range of 0.5~20 A g^−1^. (**e**) The specific capacitance of PCS, PCS-MnO_2_ composites at the current density in the range of 1~20 A g^−1^. (**f**) The areal capacitance of graphene, PCS and PCS-MnO_2_-2 at a scan rate of 1 mV s^−1^.

**Figure 6 nanomaterials-10-01097-f006:**
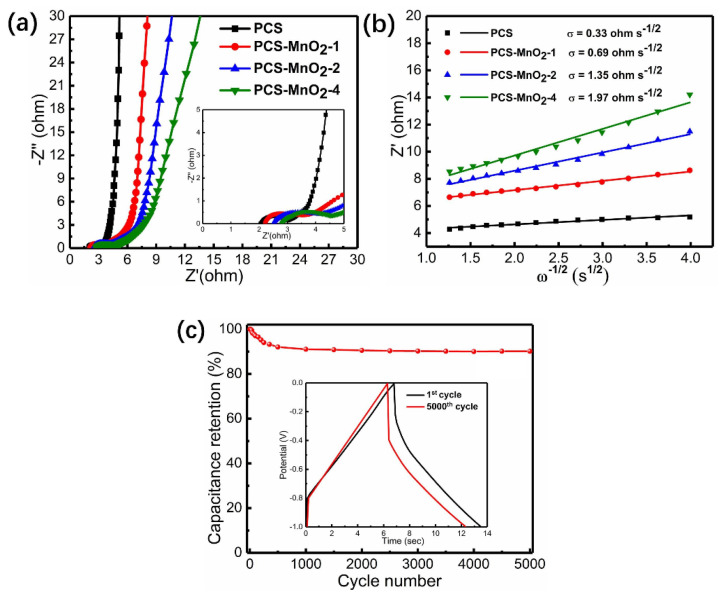
(**a**) The nyquist plots of PCS and PCS-MnO_2_-*x* (*x* = 1, 2, 4) composites (inset: magnified semicircular part) (**b**) Linear fitting to the real part of impedance (Z′) versus the −1/2 power of the angular frequency (w^−1/2^) plots in a frequency range of 0.01 Hz to 0.1 Hz to estimate the ion diffusion resistance (*σ*) of PCS and PCS-MnO_2_ composites. (**c**) The cycling performance at 10 A g^−1^ of PCS-MnO_2_-2 (inset: comparison of the first and the last charging-discharging cycle in cyclic stability test).

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
