# Peer review of "Controllable Carbonization of Plastic Waste into Three-Dimensional Porous Carbon Nanosheets by Combined Catalyst for High Performance Capacitor"

_nanomaterials, 2020, doi:10.3390/nano10061097_

Round 1

Reviewer 1 Report

The authors have substantially improved their manuscript in this revised version by presenting more data to support their conclusions such as adding the results regarding the different carbon/MnO2 ratio of the composite material, diffusion rate evaluation of the materials depending on the added MnO2 amount. However, I still have some questions and suggestions.

-One important remark is about the impedance data in Figure 5a and 5b, the authors have calculated the warburg coefficient and found a trend that warburg coefficient increases with the increased amount of MnO2. To the knowledge of the reviewer, the warburg coefficient is inversely proportional to the diffusion coeffient. Does it mean that the diffusion of ions are more difficult in the composite material? A clear comment discussing this point should be added after line 329. It would be useful to give the relationship between the warburg coefficient and the diffusion coefficient. Also, the unit of the warburg coefficient in the inside of the Figure 5b should be verified ("Hz" to "s" conversion).

-Another point is the mass loading of the electrode material is not given in the text, can be added in the exp. section.

-Also the authors in their response letter (response 1) provided a table with similar work to theirs and the performance of the resulting materials. This table can be added in the Supporting information file with a small comparative discussion in the main text. It seems that one relevant reference on the PET-based carbon from this table can be cited in the manuscript (10.1016/j.jpowsour.2009.12.090)

Line 174-175: the description of the TEM images should be modified considering that only local morphological information can be obtained from this analysis.

small typos should be corrected, 

like in Line 186: change "slightlyhigher" to "slightly higher"

Author Response

Reviewer 1

The authors have substantially improved their manuscript in this revised version by presenting more data to support their conclusions such as adding the results regarding the different carbon/MnO2 ratio of the composite material, diffusion rate evaluation of the materials depending on the added MnO2 amount. However, I still have some questions and suggestions.

1.-One important remark is about the impedance data in Figure 5a and 5b, the authors have calculated the warburg coefficient and found a trend that warburg coefficient increases with the increased amount of MnO2. To the knowledge of the reviewer, the warburg coefficient is inversely proportional to the diffusion coeffient. Does it mean that the diffusion of ions is more difficult in the composite material? A clear comment discussing this point should be added after line 329. It would be useful to give the relationship between the warburg coefficient and the diffusion coefficient. Also, the unit of the warburg coefficient in the inside of the Figure 5b should be verified ("Hz" to "s" conversion).

Answer: Thanks for the suggestion. Due to the deviation of the understanding of the unit of the abscissa, there is a mistake in the drawing calculation. I have clearly adjusted the coordinate unit and corrected it. In addition, after repeated calculations and trade-off comparisons, I think that it may be defective to select the frequency range of 1 ~ 10 Hz. Considering that ion diffusion occurs in the low-frequency straight line interval, it may be more appropriate to select the frequency interval as the low-frequency interval of 0.01 ~ 0.1 Hz. And the graph of this frequency interval is calculated and placed in the text.

The content has been corrected on line 332-337: The Warburg coefficient (s) values of PCS and PCS-MnO2 composites are 0.33 ohm s−1/2, 0.69 ohm s−1/2, 1.35 ohm s−1/2, 1.97 ohm s−1/2, respectively. It is known that the Warburg coefficient is inversely proportional to the ion diffusion coefficient. Therefore, it can be inferred from the Warburg coefficient and the change trend, which prove that abundant mesopores are beneficial for ion diffusion, and as the amount of loaded MnO2 increases, the difficulty of ion diffusion increases.

2.-Another point is the mass loading of the electrode material is not given in the text, can be added in the exp. section.

Answer: The content has been corrected on line 134: “the loading amount of the active material of the obtained electrode was approximately 4 mg cm2.”

3.-Also the authors in their response letter (response 1) provided a table with similar work to theirs and the performance of the resulting materials. This table can be added in the Supporting information file with a small comparative discussion in the main text. It seems that one relevant reference on the PET-based carbon from this table can be cited in the manuscript (10.1016/j.jpowsour.2009.12.090)

Answer: The content has been corrected on line 341-342: “the value is higher than those of PET-derived carbon materials previously reported (table S1)”, and the table has been added to the support information.

4.-Line 174-175: the description of the TEM images should be modified considering that only local morphological information can be obtained from this analysis.

Answer: The content has been corrected on Line 176-180: “It is easy to observe that the morphology of the PCS-MnO2 composite is the flake MnO2 with a diameter of about 15 nm attached to the surface of the PCS. With the increase of the amount of KMnO4 added, the amount of flake MnO2 loaded on the PCS surface gradually enhanced from PCS-MnO2-1 to PCS-MnO2-4, and the thickness of MnO2 also increased slightly.”

5-like in Line 186: change "slightlyhigher" to "slightly higher"

Answer: The content has been corrected on line 191

Reviewer 2 Report

Almost all corrections I requested have been made appropriately. As a result, the manuscript is about ready for publication, except for the following point.

Response 8: "potential plateau" is much common than "potential platform" in this case.  

Author Response

Reviewer 2

1-.Response 8: "potential plateau" is much common than "potential platform" in this case.

Answer: The content has been corrected on line 288.

Reviewer 3 Report

The revision is satisfactory.

Author Response

Thank for the reviewer's comment.

This manuscript is a resubmission of an earlier submission. The following is a list of the peer review reports and author responses from that submission.

Round 1

Reviewer 1 Report

The manuscript entitled “Controllable Carbonization of Plastic Waste to Three-Dimensional Porous Carbon Nanosheets by Combined Catalyst for High Performance Capacitor” describes the preparation of a composite material of carbon and MnO2 and their test for electrochemical supercapacitors. The examples of this type of composite materials are abundant in the literature. The interest of the work could be the use of PET and its carbonization to lead to a carbon based electrode material. The authors indicate that the carbon yield is slightly higher using a combined catalyst. However, in the end, when the carbon material is used in combination with the MnO2, the performance of the electrodes are still modest. Therefore, it is difficult to say that these composites are significantly better than various examples exist in the literature. Also, the authors only tested one ratio (carbon/MnO2 ratio) of the composite material (which was not indicated), several parameters could have been studied, especially for such a known composite. The loading, thickness of the electrodes were not provided and their effect was not tested. The electrodes before and after tests were not verified to see if they keep intact. There are some slight inconsistency in the text and the figure captions which could have been easily avoided. Based on the above comments (I have some more specific comments below), I cannot recommend the publication of the manuscript in Nanomaterials.

  • The cycling stability of the electrodes at high current density (Figure 5b) was shown but the capacitance values are very low about 50 F/g. How do the results compare to the previous work in the literature, using carbons obtained from PET but with other catalysts?
  • Equation 3: why is there a factor 100 in this formula?
  • Line 181: What do the authors mean with “microstrcture is carbon block (Figure 2a).”?
  • Normally, figure 1b and 2d are both PCS from PET using a mix catalyst, how is the difference in the morphology explained?
  • In term of morphology, it is difficult to see any difference in the figures 1d an 1e, SEM images in the absence and presence of MnO2.
  • Line 175: (Figure 1d inside red circle) probably the authors mean Figure 1e?
  • No scale bar in Figure 1e.
  • Line 163: “Notably, this curved carbon nanosheets can effectively prevent carbon sheets from re-stacking together.” What do the authors mean?
  • The authors discussed the morphological differences of the PCS in the absence/presence of MnO2: line 246 “The reduced specific surface area and pore volume of PCS-MnO2 relative to PSC implies that most of MnO2 flakes grew on the surface of the PSC. The large specific surface area and unique pore size distribution endows the PCS-MnO2 composite based electrode with fast ions diffusion for good rate capability.” Do the authors have an estimation of the diffusion coefficient of ions in both electrodes to support this idea, for example from Randles-Sevcik Equation?
  • Related to the above comment: in the interpretation of the EIS data, the authors may analyze their data more profoundly to see the differences in terms of diffusion behavior in the Warburg region. Also, the high freq. region of their spectra, there is an inductive loop, the authors have an explication for that?
  • Line 130: The authors wrote that « The density functional theory (DFT) and Brunauer-Emmett-Teller (BET) methods were used to calculate the specific surface area and pore size distribution.” The information you get from the methods is in order of “pore size distribution and specific surface area”.
  • The interpretation of the N2 sorption results should be verified and limited to what we can actually obtain from the technique. The authors indicate that there is abundant macroporosity in their material based on the isotherms. To the knowledge of the reviewer, the N2 sorption does not analyze the macroporosity, the mercury sorption experiments are considered more suitable.

Minor comments:

Line 22, abstract: remove fullstop after « up to now ».

Line 61, introduction: remove fullstop after « [17] ».

Line 66, introduction: reference style should be homogenized.

Line 82: Define “Cobalt(III) acetylacetonate” where first mentioned.

Line 86: Change “MnO2” to “MnO2

Line 88: Change “improved” to “improve”

Line 113: Change “was dissolve” to “was dissolved”.

Reviewer 2 Report

This paper describes the synthesis and capacitive properties of PCS-MnO2 composite. PCS was derived from PET. The biggest problem in this study is the interpretation of electrochemical tests. Therefore, I cannot recommend the submitted manuscript to be published in Nanomaterials at the current stage. A major revision is necessary for further consideration for publication. Especially, the paragraph for explaining Fig. 4 seems to be thoughtless. The assignment of current peaks and the rate capability must be reconsidered and well-explained.

There are many careless mistakes and grammatical errors in English writing, which reduces the value of the research. Clear examples are listed below (not exhaustively):

Line 22: “Up to now. various of strategies have” should be revised.

Line 107: “an autoclave” is correct. Are “700 ℃”, “autoclave” correct?

Line 125: What is “carbon nanoparticle”?

Line 187: It is difficult to understand the core-shell structure from Fig. 2c.

Line 189: This sentence is grammatically incorrect? The same is true in some other places.

Figure 2a: Letters in the inset are too small to see.

Line 209: “peaks typical of MnO2” and “are marked” are correct.

Line 272: “platform”?

Figure 4a: The unit A/g in the y-axis should be unified to the others.

Reviewer 3 Report

This manuscript reports preparation and characterization of MnO2 loaded porous carbon synthesized from PET with catalyst.  The effectiveness of MgO-CoO catalyst is clearly demostrated and supercapacitor performance is excellent.  I recommend publication after amendment about the following points:

  1. It is difficult to understand what kind of autoclave was used (size, material, volume of PET waste) to non-specialist.  I want to know how air-tight it is.  If the autoclave is too small, it is impossible to use this process in the real recycling of PET.
  2. Where is the lost carbon?  Did it escape from the autoclave as CO2 or hydrocarbon?
  3. Fig.4(d) is hard to see.  How about using log scale in the x-axis(time)?
  4. Figs. 4(b)(d) shows kinks.  What do they correspond to?
  5. Fig.5(b): The degradation occurs during first 200 cycles and after that, the capacity is almost constant.  It is better to show the degradation after initial decay (-> 98%?)
  6. Some errors in spelling and English:  kHZ-> kHz, various of strategies -> various strategies etc.